# Pinealectomy-Induced Melatonin Deficiency Exerts Age-Specific Effects on Sphingolipid Turnover in Rats

**DOI:** 10.3390/ijms26041694

**Published:** 2025-02-16

**Authors:** Jane Tchekalarova, Irina Georgieva, Teodora Vukova, Sonia Apostolova, Rumiana Tzoneva

**Affiliations:** 1Institute of Neurobiology, Bulgarian Academy of Sciences, Acad. G. Bonchev Street, Block 23, 1113 Sofia, Bulgaria; 2Department of Organic Chemistry, University of Chemical Technology and Metallurgy, 1756 Sofia, Bulgaria; 3Institute of Biophysics and Biomedical Engineering, Bulgarian Academy of Sciences, Acad. G. Bonchev Street, Block 21, 1113 Sofia, Bulgaria; igeorgieva@biomed.bas.bg (I.G.); tivukova@bio.bas.bg (T.V.); sonia_apostolova@yahoo.com (S.A.)

**Keywords:** pinealectomy, aging, sphingolipid turnover, ceramide, rat

## Abstract

The existing body of literature, in conjunction with our recent studies, shows that melatonin dysfunction can accelerate the aging process, with this effect depending on the specific age of the subject. The present study aims to ascertain the impact of pinealectomy on sphingolipid (SL) turnover in young adult (3-month-old), middle-aged (14-month-old), and old (18-month-old) rats. Ceramide (Cer) levels, neutral (NSMase) and acid sphingomyelinase (ASMase), acid ceramidase (ASAH1), and sphingosine-1-phosphate (S1P) levels in hippocampus and/or plasma, were evaluated by enzyme-linked immunosorbent assay. The accumulation of Cer and its metabolite second messenger S1P in the hippocampus and plasma was associated with increased levels and activity of hippocampal NSMase in the hippocampus and plasma. However, no such association was observed for hippocampal ASMase, whose levels and activity were reduced in middle-aged and old rats compared to young adult rats. Pinealectomy-induced melatonin deficiency in young adult rats showed an increase in hippocampal Cer content compared to the sham group. However, in contrast to young adult rats, pinealectomy had an inverse effect on age-related changes in hippocampal Cer, NSMase, and ASMase in middle-aged rats. Furthermore, pinealectomy exacerbated the age-related increase in S1P in the hippocampus of 18-month-old rats. Collectively, the results of the present study suggest that melatonin deficiency may influence the aging process by modulating SL turnover in an age-specific manner.

## 1. Introduction

By the year 2050, it is estimated that approximately 20–25% of the global population will be aged 60 and over, with certain regions, such as Europe and East Asia, experiencing even higher proportions. Aging is a multifaceted biological process that impacts nearly all aspects of cellular and systemic function. In the context of the human brain, aging is a significant risk factor for neurodegenerative diseases, including Alzheimer’s disease (AD), Parkinson’s disease (PD), and other forms of dementia.

The hippocampus, a brain structure that plays a critical role in learning, memory, and emotional control, may be particularly vulnerable to the effects of aging [1,2]. Aging is associated with neuronal loss in the hippocampus, reduced neurogenesis in the dentate gyrus, and increased oxidative stress (OS) and chronic neuroinflammation, which can impair hippocampal neurons. Melatonin, a hormone with antioxidant, anti-inflammatory, and circadian-regulating properties, has an important role in mitigating hippocampal aging [1,2]. The pineal gland, where melatonin is synthesized as a hormone, is often considered a key regulator of circadian rhythms and is thought to play a central role in modulating growth, fertility, aging, and death [3]. A disturbance in circadian rhythms with advancing age has been shown to reduce sensitivity to diurnal cycles in the hippocampus, which is a critical brain region for learning and memory [4]. Melatonin has been shown to influence the synchronization of the central and peripheral clocks, including the hippocampal clock, with the central circadian system. This synchronization has been demonstrated to optimize the function of the clocks. Furthermore, impaired melatonin signaling can reduce hippocampal-dependent cognitive processes, such as spatial and episodic memory. The pineal gland is believed to undergo a series of changes with advancing age, including calcification and a decline in melatonin production [5,6,7]. This decline is associated with several aging-related processes, including a decreasing antioxidant defense system, weakened immune function, and disruptions in circadian rhythms. The hypothesis that decreased melatonin levels contribute to cellular aging and the subsequent development of age-related diseases, including neurodegenerative disorders, is a compelling one.

The alterations in the sphingolipid (SL) metabolism leading to oxidative dysfunction are critical to age-related processes and diseases [8,9,10]. There are few data in the scientific literature indicating a link between the melatonin system and SL signaling [11]. SLs are a diverse group of amphipathic lipids that are abundant in cell membranes and can be classified into two main classes according to the type of polar group in their structure: phosphosphingolipids (PSLs), including sphingomyelin (SM), and glycosphingolipids (GSLs)—cerebrosides and gangliosides [12]. SLs fulfill a pivotal structural role in biological membranes, including neuronal membranes, and serve as precursors of numerous bioactive metabolites that regulate a broad array of cellular functions, including the regulation of senescence, a critical aspect of aging [13]. In this regard, the precise balance between the synthesis and degradation of SLs is typically imperative for the orchestration of numerous biological processes [8], underscoring the significant impact of changes in their metabolism on homeostatic balance and brain function.

Sphingomyelin is a significant source of ceramides (Cer), which are lipid second messengers formed when SM is cleaved by neutral or acid sphingomyelinases (NSMase and ASMase). The latter enzymes are activated by inflammatory cytokines and OS [9] or by the inhibition of Cer metabolism enzymes such as SM synthase, which converts Cer to SM, and acid ceramidase (ASAH1), which hydrolyzes Cer to sphingosine 1-phosphate (S1P) [10]. Ceramide, an important bioactive molecule, plays a regulatory role in various physiological processes, including cell proliferation, differentiation, and programmed cell death (apoptosis) [14]. The conversion of SM to Cer, induced by various stimuli, has been observed in many cell types, including neurons. This conversion results in toxicity, manifesting as pro-apoptotic effects, which can lead to tissue damage and organ dysfunction. Concurrently, the decline in cellular S1P has been demonstrated to stimulate cellular senescence [13]. Conversely, low levels of Cer and high level of S1P have trophic effects and promote survival after cell division. The differential regulation of these opposing pathways is known as the “SL Rheostat”, demonstrating Cer’s ability to induce cell growth arrest and apoptosis, while S1P is responsible for optimal cell proliferation and growth as well as the suppression of Cer-mediated growth arrest and apoptosis [15]. One of the major activators of Cer generation is OS, which is caused by reactive oxygen and nitrogen species. A link between aging tissue and NSMase activity has also been reported [16] along with a correlation between NSMase activity and cellular senescence [17]. Although ASMase is more active than NMase, higher activity of both enzymes was observed in the liver of aged rats than in young ones [18]. These enzymes are regarded as biomarkers of aging because their levels increase with age in mammals and are believed to contribute to age-related diseases, including type 2 diabetes, cardiovascular and immune dysfunction, cancer, and neurodegeneration. However, Cer may not be the sole contributing factor to the induction of senescence and aging. ASAH1, which catalyzes Cer into fatty acid and sphingosine, has also been shown to be highly upregulated in senescent cells, and silencing ASAH1 in the presence of human fibroblasts decreased the expression levels of senescent-associated proteins P53, P21, and P16 [19]. The close association between Cer accumulation, increased OS, and insulin resistance is intriguing, as these changes are thought to accelerate aging and age-related diseases. In adult rats, accumulation of specific Cer species in mitochondria has been found, a process that correlates with impaired function of complex IV in the electron transport chain [20] and corresponds to increased reactive oxygen species (ROS) during aging. A negative correlation has been observed between glutathione levels and mitochondrial NSMase activity, suggesting that the therapeutic effect of antioxidant molecules works by preventing NSMase activation and Cer accumulation by maintaining normal glutathione levels. These findings support the hypothesis that Cer accumulation contributes to age-related diseases and that strategies aimed at preventing such accumulation may potentially reduce the incidence and severity of these diseases. Melatonin has been demonstrated to inhibit ASMase in the hippocampus of mice and cell lines in a manner analogous to antidepressants, while the diurnal regulation of S1P and sphinganine 1-phosphate is dependent on intact melatonin signaling [7]. Previous publications provide a foundation for formulating a hypothesis that there is a functional link between the melatonin system and SL pathways that is critical for aging processes, including impaired homeostasis of oxidative status in the central nervous system (CNS).

The Pierpaoli et al. team reported that melatonin hormone deficiency has strict age-dependent and differential effects on lifespan and critical growth and aging parameters [21]. These authors found that pinealectomy had opposite effects on the aging process when performed on young adult mice (3 months old) and middle-aged mice (14 months old), while it had no effect on mice older than 18 months [22,23]. This observation prompted us to conduct a comparative analysis of SL metabolism in sham-operated and pinealectomized rats at 3, 14, and 18 months of age, respectively. At these ages, melatonin deficiency was found to accelerate, decelerate, or have no effect on the aging process in mice and rats [21,24,25]. The present study aims to elucidate the relationship between the pineal hormone and the SL pathway in the mechanisms mediating the aging process. The key metabolites of SM, such as Cer and S1P, along with enzymes involved in their processing, namely NSMase, ASMase, and ASAH1, were assessed at specific age periods. The findings of this study will contribute to our understanding of the relationship between oxidative status in the CNS and perturbations in SM metabolism during the aging process. In addition, this study will elucidate the role of melatonin deficiency in modulating these processes from an ontogenetic perspective. This aspect is believed to be a critical factor in accelerated aging under conditions of melatonin deficiency at certain stages of age.

## 2. Results

### 2.1. Melatonin Deficiency Associated with Hormonal Dysfunction (Pinealectomy) Can Modify Age-Related Cer Increase in the Hippocampus but Not in Plasma

Our finding indicate that aging is associated with increased levels of SM (*p* < 0.001, 18-month-old vs. 3-month-old sham rats) and Cer levels (*p* = 0.01, 18-month-old vs. 3-month-old sham rats) in the hippocampus (Figure 1 and Figure 2A). Furthermore, we observed an increase in Cer levels in the plasma of 18-month-old rats compared to 3-month-old animals (*p* = 0.042, 18-month-old vs. 3-month-old sham rats) (Figure 2B). These findings suggest that pinealectomy can modify Cer levels in a complex manner across different age stages. Specifically, while pinealectomy in young adult rats led to an increase in Cer levels in the hippocampus (*p* = 0.036, 3-month-old pinealectomized (pin) rats vs. matched sham rats), melatonin deficiency in middle-aged rats was associated with a decrease in Cer content (*p* = 0.006, 14-month-old pin rats vs. sham rats) (Figure 2A).

### 2.2. Melatonin Deficiency Reversed Age-Related Decreases in ASAH1 Levels in the Hippocampus but Not in the Plasma of 14-Month-Old Rats

ASAH1 has also been demonstrated to play a role in the regulation of Cer levels through the process of hydrolysis, which results in the production of sphingosine. A subsequent analysis revealed that there was no statistically significant variation in ASAH1 levels between the sham and pin groups in either the hippocampus or the plasma (Figure 3A,B).

### 2.3. Melatonin Deficiency Had Beneficial Effect on Age-Associated Changes of NSMase and ASMase in the Hippocampus Only in the Middle-Aged Rats

Given the hypothesis that SM catabolism may be the primary source of cellular Cer, a comprehensive investigation was conducted into NSMase levels (hippocampus) and activity (plasma), as well as ASMase levels and activity (hippocampus), in young adult, middle-aged, and old sham rats and rats with pinealectomy. The results show that old sham rats exhibited increased levels of NSMase in the hippocampus (*p* = 0.01, 18-month-old vs. 3-month-old sham rats). However, the plasma activity of the enzyme showed only a slight tendency to increase with age (Figure 4A,B). The removal of the pineal gland led to a reduction in NSMase levels to those observed in the control group in middle-aged rats (*p* < 0.001, 14-month-old pine rats vs. sham rats) (Figure 4A).

In contrast to NSMase, the content of which in the hippocampus increased with age, ASMase exhibited the opposite age-related changes. ASMase levels were decreased in middle-aged and old sham rats compared to young adult rats (*p* = 0.002 and *p* = 0.005, respectively) (Figure 5A). Consequently, the 3- and 14-month-old sham rats exhibited higher ASMase activity compared to the 18-month-old matched control (*p* < 0.001, 3- and 14-month-old vs. 18-month-old sham rats) (Figure 5B). Conversely, pinealectomy resulted in the restoration of enzyme levels in the hippocampus to the control levels in the middle-aged rats (*p* = 0.002, 14-month-old pin rats vs. matched sham rats) (Figure 5A).

### 2.4. Melatonin Deficiency Exacerbates Age-Related Increase in S1P

S1P is a product of Cer degradation by the action of ceramidases. As with its precursor, this signaling molecule was elevated in middle-aged and old rats compared to young adult rats (*p* < 0.001) (Figure 6). Pinealectomy affected only 18-month-old rats, which showed higher S1P levels than their sham-matched counterparts (*p* = 0.007).

## 3. Discussion

The present findings reveal that melatonin deficiency associated with hormonal dysfunction induces an age-specific alteration in SL metabolism in the hippocampus in rats. The functionality of neurons is intimately associated with the homeostatic balance of sphingolipids (SLs) in both animal models and human subjects. The process of aging is susceptible to aberrations in SL metabolism, consequently resulting in the manifestation of memory impairments within the hippocampus. In this study, we showed that old rats had increased levels of SM, Cer, and Cer-derivative S1P in both the hippocampus and plasma compared to young adult rats. These changes were associated with increased activity and content of NSMase, while ASMase activity and levels in the hippocampus were decreased.

An excessive accumulation of Cer and its derivative S1P, particularly in the hippocampus, has been associated with neurodegeneration, inflammation, and apoptosis, which are hallmarks of age-related neurological disorders. Dysregulation in the metabolism of Cer and SM, which are key components of SL signaling, has been shown to accelerate the aging process and trigger neurodegeneration, inflammation, and apoptosis [9]. Our findings are partially consistent with those reported by Babenko et al. [27], who observed increased Cer content and Cer/SM ratio in the hippocampus and neocortex of 24-month-old rats compared to 3-month-old rats. These changes were associated with increased NSMase, but not ASMase activity. Cellular Cer can be generated by the hydrolysis of SM catalyzed by SMases. A decrease in ASMase activity could potentially reduce the production of Cer under acidic conditions, thereby counterbalancing the amount of Cer generated by NSMase and preventing its overproduction in the aging brain. Consequently, reduced activity of ASMase may exert an indirect protective effect on the hippocampus by preventing the overproduction of Cer that is under NSMase control in aging. Conversely, activated NSMase leading to elevated Cer levels can induce apoptosis and inflammation, disrupt membrane integrity and signaling pathways critical for cellular functions [28], and can disrupt the balance between endogenous antioxidant molecules such as glutathione and hence the production of free radicals during metabolic activity [29].

Aging is associated with a decrease in glutathione (GSH) levels, leading to OS and activation of NSMase, which hydrolyzes SM to Cer [30]. Chronic inflammation and oxidative damage in the aging brain further enhance NSMase activity. Aging-associated metabolic and inflammatory stress may enhance de novo Cer synthesis from serine and palmitoyl-CoA. This pathway may become overactive in response to insulin resistance or mitochondrial dysfunction, both of which are common in aging. Therefore, the results indicate that NSMase plays a critical role in the elevation of Cer and increase in hippocampal SM content in the hippocampus of aged rats. In aged brains, OS potentiates NSMase activity and enhances this effect. Other potential mechanisms that could stimulate the accumulation of Cer and SM in the hippocampus of aged rats could be related to a disrupted (upregulated) de novo pathway for Cer synthesis due to cellular stress and chronic inflammation, which are common in the aging brain. The hippocampus is particularly sensitive to Cer-induced neurotoxicity, affecting memory and learning, and our preliminary data confirm that old rats showed decreased cognitive capacity [31]. Cer accumulation is a hallmark of aging, but increased SM levels have also been reported as a compensatory response, converting Cer back into SM [32]. Dysregulated NSMase activity in the plasma can lead to inefficient degradation of SM, contributing to its accumulation. Elevated levels of SM can stiffen cellular membranes, interfering with vesicular trafficking and receptor signaling. Excess SM may compete with other phospholipids, disrupting lipid homeostasis and membrane integrity. Finally, elevated levels of total brain SM are a pivotal pathological occurrence that contributes significantly to neurodegeneration [33].

A review of the relevant literature and our previous data indicate that pinealectomy is associated with various neurobiological changes, particularly in the hippocampus, which plays a central role in memory, learning, and emotional regulation [21,24,25,26]. Surgical removal of the pineal gland leads to significant changes in the brain, particularly in the hippocampus, due to the loss of melatonin production [34]. However, the present findings show that pinealectomy-related changes in the hippocampus do not always correspond to alterations in plasma. Specifically, the results indicate that age-related deterioration of serotonin (5-HT) turnover can be exacerbated by pinealectomy-induced melatonin deficiency in middle-aged rats with overproduction of Cer in the hippocampus but not in the plasma, suggesting an increased vulnerability of this brain region. Furthermore, this effect was not associated with a proportional change in NSMase (increased level) or ASMase (decreased activity and/or level). This discrepancy could be explained by the fact that NSMase levels, but not activity, were measured in the hippocampus, which is a limiting factor of this study. In addition, endogenous melatonin has been shown to be a potent scavenger of OS-related molecules [35]. In this regard, we previously reported that pinealectomy increased lipid peroxidation in the hippocampus of 3-month-old rats and decreased superoxide dismutase (SOD) activity during the dark phase compared with the corresponding sham rats [26]. The elevated OS observed in the present study may be due to the increased Cer production.

The present study found that the level of Cer was decreased in the hippocampus of middle-aged rats with pinealectomy compared with the corresponding sham group. In addition, while the level of NSMase was also reduced, the level of ASMase was restored in the hippocampus of 14-month-old pinealectomized rats compared with sham rats of the same age. This finding was unexpected, given our previous report that this age group showed the greatest susceptibility to OS in the hippocampus following pinealectomy [26]. Consistent with the findings in young adult rats, pinealectomy decreased the heat shock protein 70 expression in the frontal cortex, impaired spatial working memory, and reduced the expression of the brain-derived neurotrophic factor (BDNF) and its receptor, TrkB [31], and liver enzymes (alanine aminotransferase [ALAT] and aspartate aminotransferase [ASAT]) [26] in middle-aged rats. Furthermore, we recently reported that, at this particular stage, there was a decrease in SM levels in the hippocampus of rats following pinealectomy, as compared to the matched control group [26]. Significantly, we also found that melatonin deficiency in 14-month-old rats is associated with a disturbance in the antioxidant/oxidant balance within the rat hippocampus, characterized by decreased GSH and increased lipid peroxidation. It is noteworthy that young adult and old rats with pinealectomy did not show these changes. These findings suggest that endogenous melatonin in mature rats plays a critical role in mitigating free radical production by promoting GSH production. In addition, research has shown that OS-induced changes in the hippocampus result in decreased SM levels within the same structure. Previous studies suggested that the antidepressant properties of melatonin are associated with suppression of Cer accumulation, possibly through inhibition of SM metabolism [7].

## 4. Materials and Methods

The experimental protocol was conducted in agreement with the European Communities Council Directive 2010/63/E.U. and approved by the Bulgarian Food Safety Agency (research project: #300/N◦5888–0183).

### 4.1. Animals

Male Wistar rats of three different ages were purchased from the vivarium of the Institute of Neurobiology, Bulgarian Academy of Sciences. Animals were housed for at least one week prior to surgery in standard Plexiglas cages (n = 3–4) under a 12 h/12 h light/dark regime, light on at 07:00 AM, room temperature at 22–23 °C, with ad libitum access to food and water.

### 4.2. Experimental Groups and Surgery

Six experimental groups were assigned according to age and surgical procedure as follows: 3-month-old sham group; 3-month-old pin group; 14-month-old sham, 14-month-old pin, 18-month-old sham, and 18-month-old pin groups, respectively, n = 8. The pinealectomy or sham procedure (the same procedure except for the last step of the pinealectomy) was performed according to the protocol described in our previous studies [25]. Briefly, anesthetized rats (mixture of ketamine, 40 mg/kg, i.p., and xylazine, 20 mg/kg, i.m. for the 3-month-old rats; ketamine, 60 mg/kg, and xylazine, 30 mg/kg, i.m. for the 14-month-old rats) were fixed on a stereotaxic apparatus (Stoelting, Wood Dale, IL, USA). After craniotomy with a small dental drill, the pineal gland located under the venous sinus was quickly removed with fine forceps. The rats were treated with antibiotic injection for three days after surgery. At least 2 weeks after the surgical manipulation, the animals were euthanized with a guillotine after light anesthesia with CO_2_, the two hippocampi were briefly removed, snap frozen in liquid nitrogen and stored at −20 °C until further analysis (6–8 samples/group), and their trunk blood was collected. The experimental design is shown in Figure 7.

### 4.3. ELISA

#### 4.3.1. Ceramide Levels in Hippocampus

Hippocampal tissue was weighted and homogenized (BANDELIN electronic GmbH&Co. KG, Berlin, Germany) in ice-cold PBS to a final concentration of 60 mg/mL. Then, the homogenates were centrifuged for 15 min, 1500× *g*, 4 °C, and the supernatant was collected and assayed immediately or aliquot and stored at −80 °C. The samples were processed according to a Rat Ceramide ELISA kit (Cat. No. E02C2522, BlueGene Biotech, Shanghai, China) without further dilution. They were measured at 450 nm in a microplate reader (Tecan Infinite F200 PRO (Tecan Trading GmbH, Männedorf, Switzerland) and the samples’ concentrations were calculated based on a standard curve at ng/mL.

Hippocampal tissue was weighted and homogenized (BANDELIN electronic GmbH&Co. KG, Berlin, Germany) in ice-cold PBS containing a protease inhibitor cocktai, to a final concentration of 60 mg/mL. Then, the homogenates were centrifuged for 15 min, 1500× *g*, 4 °C, and the supernatant was collected and assayed immediately or aliquot and stored at −80 °C. The samples were processed according to a Rat Ceramide ELISA kit (Cat. No. E02C2522, BlueGene Biotech, Shanghai, China) without further dilution. The amounts of ceramide levels were calculated based on a standard curve at ng/mL after measuring the OD of the samples at 450 nm in a microplate reader (Tecan Infinite F200 PRO (Tecan Trading GmbH, Männedorf, Switzerland).

#### 4.3.2. Ceramide Levels in Blood Plasma

The serum, obtained from the blood samples of the animals, was clotted for 10–20 min (room temperature), while the plasma was collected in EDTA tubes and mixed for 20 min. Then, the samples were centrifuged for 20 min at 2000–3000 rpm, and the supernatants were collected and stored at −20 °C until assayed. Blood ceramide levels (mmol/L) were determined via a commercial ELISA kit (Cat. No. MBS3809105, MyBioSource, Inc., San Diego, CA, USA). The samples’ OD was measured at 450 nm using a microplate reader (Tecan Infinite F200 PRO (Tecan Trading GmbH, Männedorf, Switzerland).

#### 4.3.3. Sphingomyelin Levels in Hippocampus

Rat hippocampal samples were homogenized to a final concentration of 100 mg/mL in cell/tissue extraction buffer (100 mM Tris, pH 7.4; 150 mM NaCl, 1 mM EGTA, 1 mM EDTA, 1% Triton X-100, 0.5% Na deoxycholate, protease inhibitor cocktail). After centrifugation at 4 °C and 10,000× *g* for 20 min, the supernatants were transferred to separate tubes and were heated for 5 min at 70 °C until they became cloudy. To remove any debris, the samples were centrifuged again at 10,000× *g*, 2 min. Then, 100 μL of each supernatant was transferred in duplicates to a 96-well plate. The SM concentration was estimated using an SM kit according to the manufacturer’s instructions (Cat No. MBS265875, MyBioSource, Inc., San Diego, CA, USA). SM concentration was measured by detecting the absorbance at 450 nm using a microplate reader (Tecan Infinite F200 PRO (Tecan Trading GmbH, Männedorf, Switzerland), calculated based on a standard curve, and expressed as ng/mL.

#### 4.3.4. Neutral Sphingomyelinase Levels in Hippocampus

The collected rat hippocampal samples were weighted and homogenized (1:10) in ice-cold PBS as described above. Then, homogenates were centrifuged accordingly to obtain the supernatants. The samples were then assayed via Rat Neutral Sphingomyelinase (NSMase) ELISA kit (Cat.No. MBS452865, MyBioSource, Inc., San Diego, CA, USA) according to the manufacturer’s instructions. The samples were measured at 450 nm on a microplate reader (Tecan Infinite F200 PRO (Tecan Trading GmbH, Männedorf, Switzerland) and the obtained levels of neutral sphingomyelinase were expressed in ng/mL.

#### 4.3.5. Neutral Sphingomyelinase Enzymatic Activity in Blood Plasma

The plasma was collected in EDTA tubes and mixed for 20 min. Then, the samples were centrifuged for 20 min at 2000–3000 rpm and the supernatants were collected and stored at −20 °C until assayed. To measure the NSMase enzymatic activity, the samples were processed via an ELISA kit (Cat. No. ab138876, Abcam Limited, Cambridge, UK). The kit uses an indicator to indirectly measure the amount of phosphocholine produced during the hydrolysis of sphingomyelin by NSMase. The samples’ OD was monitored at 655 nm on a microplate reader (Tecan Infinite F200 PRO (Tecan Trading GmbH, Männedorf, Switzerland) and expressed in mU/mL.

#### 4.3.6. Acid Sphingomyelinase Enzymatic Activity in Hippocampus

The samples were assayed via Rat Acid Sphingomyelinase (ASMase) ELISA kit (MyBiosource, Cat. No. MBS017899, Inc., San Diego, CA, USA). The samples were measured at 450 nm on a microplate reader (Tecan Infinite F200 PRO (Tecan Trading GmbH, Männedorf, Switzerland) and obtained acid sphingomyelinase levels were expressed in ng/mL.

#### 4.3.7. Acid Sphingomyelinase Levels in Hippocampus

The samples were then assayed via Rat Acid Sphingomyelinase (ASMase) ELISA kit (MyBiosource, Cat. No. MBS017899, Inc., San Diego, CA, USA) according to the protocol. The samples were measured at 450 nm on a microplate reader (Tecan Infinite F200 PRO (Tecan Trading GmbH, Männedorf, Switzerland) and expressed in ng/mL.

#### 4.3.8. Acid Ceramidase Levels in Hippocampus

The samples were then assayed via Rat Acid Ceramidase (ASAH1) ELISA kit (Cat.No. abx502659, Abbexa LTD., Cambridge, UK) according to the manufacturer’s instructions. The levels of acid ceramidase were expressed in ng/mL after measuring of samples at 450 nm on a microplate reader (Tecan Infinite F200 PRO (Tecan Trading GmbH, Männedorf, Switzerland).

#### 4.3.9. Acid Ceramidase Levels in Blood Plasma

To determine plasma acid ceramidase (ASAH1) levels, a commercial Rat Acid Ceramidase ELISA kit (Cat. No. MBS283260, MyBioSource, Inc., San Diego, CA, USA) was used. The samples’ OD was read at 450 nm using a microplate reader (Tecan Infinite F200 PRO) (Tecan Trading GmbH, Männedorf, Switzerland), from which the concentration of ASAH1 was calculated in ng/mL.

#### 4.3.10. Sphingosine-1-Phosphate Levels in Hippocampus

Sphingosine-1-Phosphate (S1P) levels were measured by a competitive ELISA assay according to manufacturer’s instructions (Cat. No. abx251939, Abbexa LTD, Cambridge, UK). The OD of supernatants was measured spectrophotometrically at 450 nm using a microplate reader (Tecan Infinite F200 PRO (Tecan Trading GmbH, Männedorf, Switzerland), from which the concentration of S1P was calculated in ng/mL.

### 4.4. Statistical Analysis

Two-way analysis of variance (ANOVA) with the factors age (3 levels) and surgery (2 levels) was performed. A post hoc Bonferroni test was applied in case of significance of some of the factors. Data are presented as mean ± S.E.M. Statistical analysis was performed using SigmaStat^®^ 11.0 (Systat Software, San Jose, CA, USA) and GraphPad Prism^®^6 (GraphPad Software, San Diego, CA, USA). Significant difference was accepted at *p* ≤ 0.05.

## 5. Conclusions

Taken together, the results of the present study suggest that melatonin deficiency associated with hormonal dysfunction affects SL signaling in the hippocampus in an age-specific manner. The age-related increased Cer level was comparable to that in young adult pinealectomized rats in the hippocampus, but not in the plasma, suggesting that this brain region may have distinct mechanisms for lipid homeostasis that may be more vulnerable at this age stage under conditions of melatonin deficiency. In contrast, the reduced susceptibility of 14-month-old pinealectomized rats to age-associated impairment of hippocampal SL turnover may be not related with previously reported increased hippocampal OS [26], but to other compensatory mechanisms that need to be evaluated in the future. The limiting factor of this study is that the signaling molecules related to SL metabolism were evaluated only by ELISA, and further analyses with other methodological approaches, including immunohistochemistry, are needed to confirm the critical role of the hippocampus as well as other regions of the limbic system in pinealectomy-related changes in SL turnover at specific age stages. In addition, the effect of melatonin supplementation in melatonin deficiency at different ages will answer the question as to whether this system is directly involved in the observed phenomena or whether this condition triggers changes in other systems involved in altered SL metabolism.

## Figures and Tables

**Figure 1 ijms-26-01694-f001:**
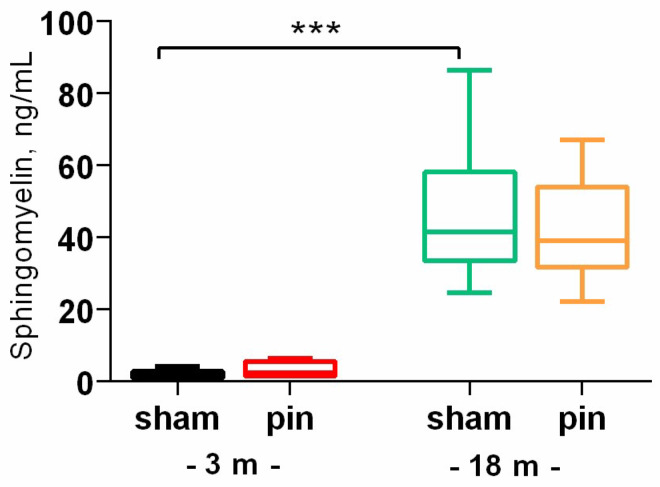
The effect of pinealectomy on sphigomielin (SM) content in the hippocampus in 3- and 18-month-old rats. Data are presented as mean ± SEM. Two-way ANOVA indicated a major age effect [F1,23 = 60.919, *p* < 0.001]; *** *p* < 0.001, 18- vs. 3-month-old sham rats. The 3 m results are published in [26].

**Figure 2 ijms-26-01694-f002:**
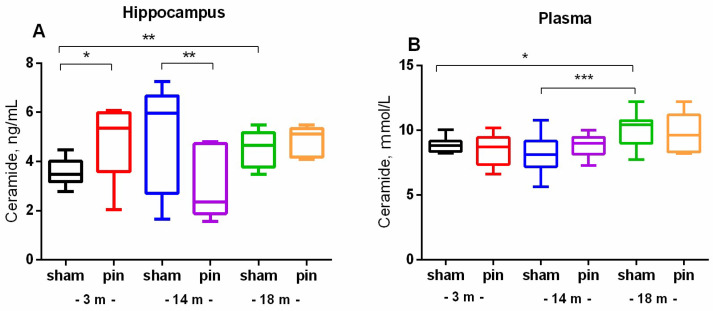
The effect of pinealectomy on ceramide (Cer) content in the hippocampus and plasma in 3-, 14-, and 18-month-old rats. Data are presented as mean ± SEM. Two-way ANOVA indicated age × surgery interaction [F2,45 = 6.915, *p* = 0.003] for the hippocampus; * *p* = 0.01, 18- vs. 3-month-old sham rats; * *p* = 0.036, 3-month-old pin vs sham rats; and ** *p* = 0.006, 14-month-old pin vs sham rats (**A**). For plasma, a major age effect [F2,52 = 3.217, *p* = 0.049] was detected; * *p* = 0.042, 18- vs. 3-month-old sham rats; and *** *p* < 001, 18- vs. 14-month-old sham rats (**B**).

**Figure 3 ijms-26-01694-f003:**
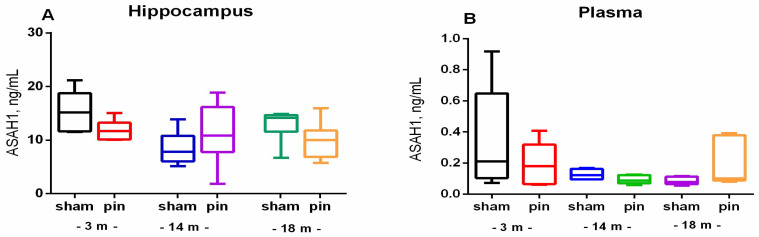
The effect of pinealectomy on acid ceramidase (ASAH1) content in the hippocampus and plasma in 3-, 14-, and 18-month-old rats. Data are presented as mean ± SEM. Two-way ANOVA did not indicate a significant effect of both age and surgery for ASAH1 content in the hippocampus and plasma (*p* > 0.05) (**A**,**B**).

**Figure 4 ijms-26-01694-f004:**
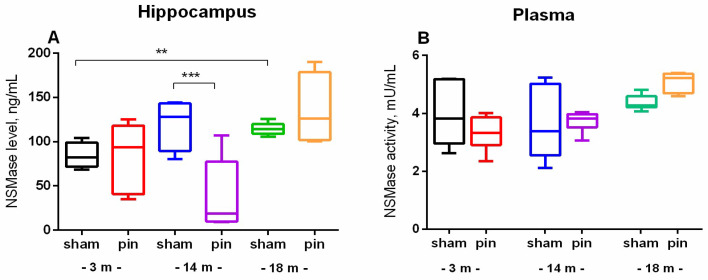
The effect of pinealectomy on NSMase levels in the hippocampus and NSMase activity in plasma in 3-, 14-, and 18-month-old rats. Data are presented as mean ± SEM. Two-way ANOVA indicated a main age effect: [F2,35 = 5.496, *p* = 0.012] as well as age × surgery interaction: [F2,35 = 6.267, *p* = 0.007] for the NSMase level in the hippocampus; ** *p* = 0.01, 18-month-old sham vs. 3-month-old sham; and *** *p* < 0.001, 14-month-old pin vs sham rats (**A**). Two-way ANOVA did not indicate a significant effect of both age and surgery for the NSMase level in plasma (*p* > 0.05) (**B**).

**Figure 5 ijms-26-01694-f005:**
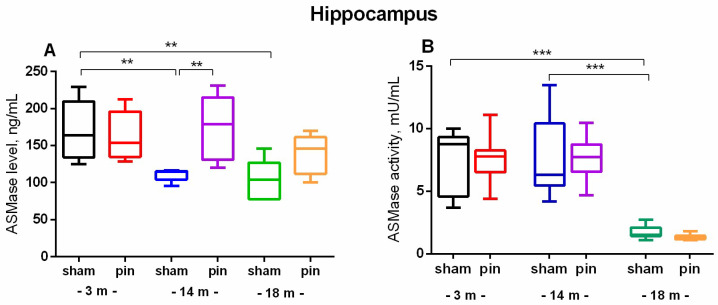
The effect of pinealectomy on ASMase level (**A**) and activity (**B**) in the hippocampus in 3, 14-, and 18-month-old rats. Data are presented as mean ± SEM. Two-way ANOVA indicated a main age effect [F2,35 = 6.584, *p* = 0.003], surgery effect [F1,46 = 7.223, *p* = 0.010] for ASMase level; ** *p* = 0.002, 14-month-old sham rats vs. 3-month-old sham rats; and ** *p* = 0.005, 18-month-old sham rats vs. 3-month-old sham rats (**A**). Two-way ANOVA showed a major age effect for the ASMase activity: F2,45 = 647.244, *p* < 0.001], and *** *p* < 0.001, 18-month-old sham rats vs. 3- and 14-month-old sham rats (**B**).

**Figure 6 ijms-26-01694-f006:**
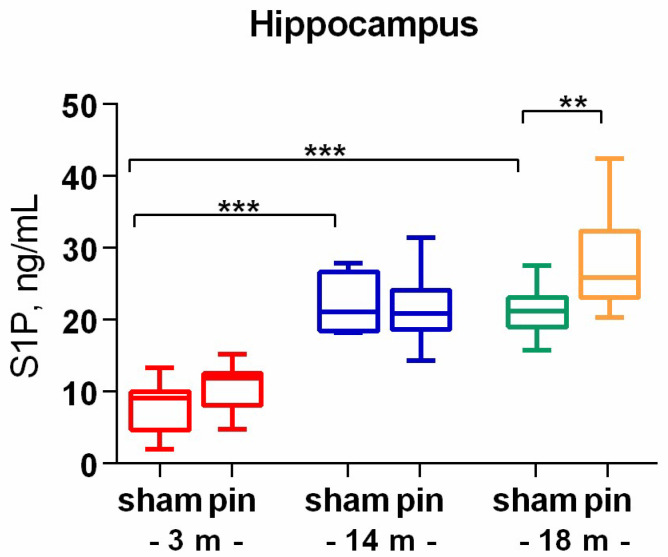
The effect of pinealectomy on Sphingosine-1-phosphate (S1P) level in the hippocampus in 3-, 14-, and 18-month-old rats. Data are presented as mean ± SEM. Two-way ANOVA indicated a main age effect [F2,35 = 47.73, *p* < 0.001], surgery effect [F1,35 = 4.99, *p* = 0.05], *** *p* < 0.001, 14- and 18-month-old sham rats vs. 3-month-old sham rats; and ** *p* = 0.007, 18-month-old pin rats vs. 18-month sham rats.

**Figure 7 ijms-26-01694-f007:**
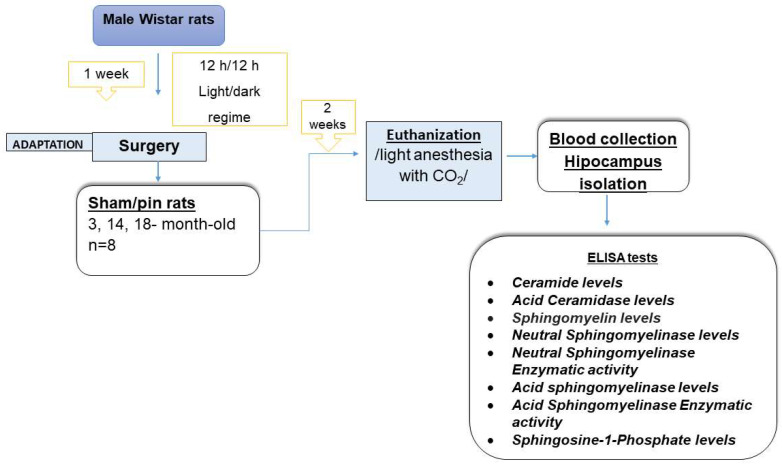
Timeline of the experimental protocol.

## Data Availability

All data generated or analyzed during this study are included in this article. Further inquiries can be directed to the corresponding authors.

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
