# Peer review of "Pinealectomy-Induced Melatonin Deficiency Exerts Age-Specific Effects on Sphingolipid Turnover in Rats"

_ijms, 2025, doi:10.3390/ijms26041694_

Round 1

Reviewer 1 Report

Comments and Suggestions for Authors

“Pinealectomy-induced melatonin deficiency exerts age-specific effects on sphingolipid turnover in young adult, middle-aged, and old rats” aimed to determine the effect of pinealectomy on sphingolipid turnover in young adult (3-month-old), middle-aged (14-month-old), and old (18-month-old) rats. The subject is very interesting, while the manuscript is not well structured. I have the following concerns:

1. The Introduction fails to clearly explain why the hippocampus was chosen as the target brain region instead of other brain regions. The authors need to elucidate the relationship between "melatonin," "hippocampus," and "aging."

2. "Materials and Methods" need to be presented before "Results".

3. Why did the author test both "hippocampus" and "plasma" for the same indicator?

4. Figure 1, when comparing the effects of age and hormonal dysfunction (pinealectomy) on sphingomyelin (SM), why were only the data from 3-month-old and 18-month-old rats compared, and why were only the data from the hippocampus compared?

5. In Figures 5 and 6, why were only the data from the hippocampus compared for ASMase level and activity, as well as Sphingosine-1-phosphate (S1P) level?

6. Due to the inconsistent changes observed in the same indicator between the "hippocampus" and "plasma," in the Discussion, the author should separately explain the mechanisms underlying the different changes induced by age and pinealectomy in both the "hippocampus" and "plasma," and also elucidate the mechanisms accounting for these distinct variations.

Author Response

Thank you for the careful evaluation of our manuscript. We have revised the manuscript taking into account the suggested modifications. All changes in the MS are highlighted with yellow background.

Point #1: The Introduction fails to clearly explain why the hippocampus was chosen as the target brain region instead of other brain regions. The authors need to elucidate the relationship between "melatonin," "hippocampus," and "aging."

Response: We appreciate this suggestion. The Introduction has been corrected and an additional paragraph has been added discussing the close relationship between the hippocampus, aging, and melatonin based on literature data (second paragraph of the Introduction).

Point #2: "Materials and Methods" need to be presented before "Results".

Response: We followed the journal's instructions for article organization. According to the guidelines, the Methods section must follow the Discussion.

Point #3: Why did the author test both "hippocampus" and "plasma" for the same indicator?

Response: The idea was to look for effects in the central nervous system (hippocampus) and the periphery (plasma). Our data suggest that the hippocampus, but not plasma, is the critical region for pinealectomy-induced changes in ceramide and related to its metabolism enzymes such as NSMase and aSMAase in middle-aged rats. This issue was discussed now in the Discussion section (line 261).

Point #4: Figure 1, when comparing the effects of age and hormonal dysfunction (pinealectomy) on sphingomyelin (SM), why were only the data from 3-month-old and 18-month-old rats compared, and why were only the data from the hippocampus compared?

Response: In Discussion, we cited our previous data on the effect of pinealectomy on SM in middle-aged rats (line 387).

Point #5: In Figures 5 and 6, why were only the data from the hippocampus compared for ASMase level and activity, as well as Sphingosine-1-phosphate (S1P) level?

Response: Ceramide is a major component of sphingolipids and an important signaling molecule that regulates various cellular processes, including cell growth, differentiation, and apoptosis. Considering that SM degradation as the main source of ceramide in cells (Gon˜ i &Alonso, 2002), we investigated the levels and activities of neutral and acid SMases that regulate ceramide levels in the hippocampus. Fibrillar Ab-activated astroglia have been shown to kill neurons via NSMase but not ASMase (Jana & Pahan, 2010). Guided by this fact, we emphasized our study of hippocampal and plasma NSMase levels and activity because of the possibility that increased levels of the enzyme in the hippocampus would lead to increased levels in plasma and other brain structures. Following the negative results obtained for the effect of aging and pinealectomy in plasma obtained with NSMase and ASAH1, and considering the more important role of NSMase than ASAH1 (the Discussion section), we evaluated the effects of ASMase only in the hippocampus.  Because ASMase plays a minor role in age-related ceamide production, we limited our study to the levels and activity of this enzyme in the hippocampal brain structure. Indeed, our results showed that the levels and activity of ASMase were decreased in middle-aged and old sham rats compared to young adult rats. Decreased activity of ASMase may be considered as a mechanism to protect the hippocampus from overproduction of ceramides by NSMase in old age.

The same is true for S1P. Since this bioactive lipid, produced exclusively by the breakdown of ceramide, is the most abundant in brain structures and is first catabolized from ceramide and sphingosine in the different cytoplasmic compartments of hippocampal neurons, we focused our study on the levels of S1P in the hippocampus. Given the important role of S1P for a stable and functional vasculature (van Echten-Deckert, 2023), we plan to investigate S1P levels in blood plasma in the future.

Point #6: Due to the inconsistent changes observed in the same indicator between the "hippocampus" and "plasma," in the Discussion, the author should separately explain the mechanisms underlying the different changes induced by age and pinealectomy in both the "hippocampus" and "plasma," and also elucidate the mechanisms accounting for these distinct variations.

Response: Following this recommendation of the reviewer, we have included inconsistent changes for certain parameters in the hippocampus and plasma in the Discussion section (line 261).

Reviewer 2 Report

Comments and Suggestions for Authors

Lines 34-40. I encourage the authors to modify this information due to its high level of similarity with the following document (https://doi.org/10.1016/j.physbeh.2022.113786).

I also encourage the authors to rephrase the following information (sections 4.3.4, 4.3.7, and 4.3.10), as they present similarities with the following reference: https://doi.org/10.3390/ijms23052809.

The authors should state the year the ethical committee approved the protocol. The "Institutional Review Board Statement" section states that the protocol was from 2021.  Was the same protocol number used for different articles?

The term decapitation should be modified for euthanized.

Lines 320-322 should be rewritten. This sentence presents a challenge to understand.

Some sections need editing for grammar and spelling.

This is a continuity study. The authors should explain in the discussion the differences between this study and the previous works. Why did they present different studies with similar methodology? Did they split the information into different manuscripts?

Lines 212 and 241. What does old mean? The age should be more specific.

There are five references from Jane Tchekalarova. Are all the references necessary for the study?

The discussion needs improvement. What are the limitations of this study? Is there another variable considered in further studies using the same animal model?

Comments on the Quality of English Language

Lines 320-322 should be rewritten. This sentence presents a challenge to understand.

Some sections need editing for grammar and spelling.

Author Response

Thank you for the careful evaluation of our manuscript. We have revised the manuscript taking into account the suggested modifications. All changes in the MS are highlighted in yellow.

Point#1: Lines 34-40. I encourage the authors to modify this information due to its high level of similarity with the following document (https://doi.org/10.1016/j.physbeh.2022.113786).

I also encourage the authors to rephrase the following information (sections 4.3.4, 4.3.7, and 4.3.10), as they present similarities with the following reference: https://doi.org/10.3390/ijms23052809.

Response: We appreciate this suggestion. The first paragraph of the Introduction has been corrected and the above-mentioned sentences were removed and replaced with other paragraph.

We rephrase the content of information mentioned by the reviewer in the sections 4.3.4, 4.3.7, and 4.3.10 of the manuscript.

Point#2: The authors should state the year the ethical committee approved the protocol. The "Institutional Review Board Statement" section states that the protocol was from 2021.  Was the same protocol number used for different articles?

Response: We cited the approved by Bulgarian Food Safety Agency permit No. 300/Nâ—¦5888–0183/10.05.2021. This permission is for several different experiments on this topic and the experimental design of this study is involved in it.

Point#3: The term decapitation should be modified for euthanized.

Response: Corrected.

Point#4: Lines 320-322 should be rewritten. This sentence presents a challenge to understand.

Some sections need editing for grammar and spelling.

Response: We agree with this suggestion of the Reviewer and corrected the above-mentioned sentence. In addition, a thorough edition of  grammar and spelling errors have been made in the text. Corrections are given in yellow.

Point#5: This is a continuity study. The authors should explain in the discussion the differences between this study and the previous works. Why did they present different studies with similar methodology? Did they split the information into different manuscripts?

Response: The results of this manuscript with a focus of age-related melatonin deficiency on SL turnover are discussed in comparison with our previous reports for the effects on oxidative stress, behavioral responses and some crucial for aging signaling molecules. We used different behavioral tests for anxiety, depressive-like behavior, motor activity, physiological parameters (blood pressure, metabolic parameters, muscle strength), different signaling molecules studied mostly in brain tissue by immunohistochemistry or biochemical assays.

Point#6: Lines 212 and 241. What does old mean? The age should be more specific.

Response: The three age stages that have been chosen in our study 3-, 14- and 18-month-old rats correspond to young adult, middle-aged and old humans.  The choice to use this specific age stage for study the role of pinealectomy is explained in the new version of the mns (The Introduction, line 117).

Point#7: There are five references from Jane Tchekalarova. Are all the references necessary for the study?

Response: As the reviewer mentioned in point #5, this study is part of an ongoing work in our laboratory for several years in the framework of several projects addressing the topic from different aspects. We have used different cohorts of rats for each experimental design to solve different problems and with specific approaches.

Point#8: The discussion needs improvement. What are the limitations of this study? Is there another variable considered in further studies using the same animal model?

Response: The experimental design consisted of the signaling molecules closely related to the levels of Cer (precursor, metabolite, enzymes) either in the hippocampus or plasma. The limitation of the study is discussed in the Conclusion in the new version.

Point#9: Comments on the Quality of English Language

Lines 320-322 should be rewritten. This sentence presents a challenge to understand.

Some sections need editing for grammar and spelling.

Response: We are thankful for this advice and carefully checked again the whole text for gramma errors.

Reviewer 3 Report

Comments and Suggestions for Authors

The manuscript titled "Pinealectomy-induced melatonin deficiency exerts age-specific effects on sphingolipid turnover in young adult, middle-aged, and old rats" provides finding into age-related changes in lipid metabolism and how melatonin may influence sphingolipid turnover. Overall, I think the study is well-conducted, there are some concerns could be improved.

1. I suggest the author could connect the concept of sphingolipid metabolism to aging-related diseases (such as neurodegeneration, cardiovascular issues).

2. The study design is generally well-executed, but further detail is needed on the rationale for the age groups selected (young adult, middle-aged, old rats). Why were these specific age ranges chosen, and what relevance do they have to the physiological processes being studied?

3. The authors should provide more details on the specific methods used to measure sphingolipid turnover.

4. How does pinealectomy-induced melatonin deficiency compare to other models of melatonin depletion in terms of relevance to human physiology?

5.  Were the rats randomized into treatment groups, and was the experiment blinded?

6. The study uses a time-point-based design, I suggest a more detailed rationale for why certain age groups were assessed at specific time points post-pinealectomy. Were there any age-related differences in the timing of effects?

7. Are the age-specific changes in sphingolipid turnover linked to alterations in melatonin receptor expression or circadian clock function?

8. How might age-specific changes in sphingolipid turnover contribute to these diseases?

9. A timeline, flowchart or diagram explaining the experimental design should be provided.

Author Response

Point#1: I suggest the author could connect the concept of sphingolipid metabolism to aging-related diseases (such as neurodegeneration, cardiovascular issues).

Response: We are thankful for this suggestion of the Rеviewer. The impact of SL metabolism to aging-related diseases was discussed both in the Introduction (from line 73) and the Discussion (from line 229).

Point#2: The study design is generally well-executed, but further detail is needed on the rationale for the age groups selected (young adult, middle-aged, old rats). Why were these specific age ranges chosen, and what relevance do they have to the physiological processes being studied?

Response: We agree with this relevant remark of the Reviewer. The text explained the choice to use these age stages was added in the last paragraph of the Introduction in the new version of the manuscript (from line 118).

Point#3: The authors should provide more details on the specific methods used to measure sphingolipid turnover.

Response: Details  for methodology used to study SL turnover are presented in the last section “Materials and Methods”. We rephrase and added additional information where was missing about the methods used to measure sphingolipid turnover.

Point#4: How does pinealectomy-induced melatonin deficiency compare to other models of melatonin depletion in terms of relevance to human physiology?

Response: This issue is crucial for the translational impact on human physiology of the present study as well as other reports using melatonin deficiency models. We have the ambition to collect the data from our previous and current studies on this topic and prepare a review, emphasizing the relevance of these models to human biochemistry and physiology.

Point#5: Were the rats randomized into treatment groups, and was the experiment blinded?

Response: The rats were first divided into three different age groups (3-, 14-, and 18-month-old rats). Each cohort was then randomized into two main groups depending on the surgical procedure (sham or pin). Furthermore, the ELISA experiments were blinded to the treatment procedure.

Point#6: The study uses a time-point-based design, I suggest a more detailed rationale for why certain age groups were assessed at specific time points post-pinealectomy. Were there any age-related differences in the timing of effects?

Response: We inserted a text in the Introduction with the rationale to use these specific age stages for pinealectomy in the present study (the last paragraph).

Point#7: Are the age-specific changes in sphingolipid turnover linked to alterations in melatonin receptor expression or circadian clock function?

Response: MT receptor expression was not evaluated in this study. However, we have preliminary data suggesting that MT1 and MT1b receptor expression was not altered as a result of melatonin deficiency.

Point#8: How might age-specific changes in sphingolipid turnover contribute to these diseases?

Response:  As mentioned in the response to point #4, we have an idea to summarize and discuss the effects of melatonin deficiency at different ages on physiology and behavior and the underlying signaling pathways, including the role of changes in SL turnover.

Point#9: A timeline, flowchart or diagram explaining the experimental design should be provided.

Response:  We are thankful for this suggestion. In the new version of the manuscript a figure with experimental schedule was added to the text. 

Reviewer 4 Report

Comments and Suggestions for Authors

This study revealed that melatonin deficiency modulates aging by altering sphingolipid turnover in an age-specific manner. However, the author needs to address several crucial issues in his analysis and conclusion.

1. This data is insufficient solid evidence to support the author’s conclusion. The author needs other tools and evidence to support it. 

2. Line 141-142, the author needs to correct format. 

3. Line 149, "***p < 001, 18- vs 14-month-old sham rats." The error needs to be corrected. 

Author Response

Reviewer #4

Point#1: This data is insufficient solid evidence to support the author’s conclusion. The author needs other tools and evidence to support it. 

Response:  We agree with this remark of the Reviewer. The limitations of the study are discussed in the Conclusion and the need for further analysis of the impact of this model and other with melatonin deficiency on SL turnover in the CNS.

Point#2: Line 141-142, the author needs to correct format. 

Response:  The format was again carefully checked and corrected.

Point#2: Line 149, "***p < 001, 18- vs 14-month-old sham rats." The error needs to be corrected. 

Response:  We are thankful for this note. The missing asterisks were inserted in the text to the Fig. 1.

Round 2

Reviewer 2 Report

Comments and Suggestions for Authors

The manuscript can be accepted for publication

Author Response

We thank you for all the comments and recommendations on our manuscript, which will help improve its quality.

Reviewer 3 Report

Comments and Suggestions for Authors

No further comments.

Author Response

We thank you for all the comments and recommendations on our manuscript that will help to improve its quality.

Reviewer 4 Report

Comments and Suggestions for Authors

The author needs to provide sufficient solid evidence to support the author's conclusion. 

Author Response

Reviewer #4

Point #1 The author needs to provide sufficient solid evidence to support the author's conclusion. 

Response: Considering the reviewer's recommendation, the Conclusion was changed. Only the first sentence from the first version of the mns was included in the second version. Please see the corrected Conclusion (in red) (line 458-line 453).

Round 3

Reviewer 4 Report

Comments and Suggestions for Authors

The author need to improve it. 

Author Response

The text was corrected according to the notes of the reviewer and academic reviewer. All corrections in the new version are highlighted in grey. 
